# An Ensemble Model for Multi-Level Speech Emotion Recognition

**Chunjun Zheng [1,2,\*], Chunli Wang [1] and Ning Jia [2]**

1   College of Information Science and Technology, Dalian Maritime University, Dalian 116026, China; clwang@dlmu.edu.cn
2   School of Computer and Software, Dalian Neusoft University of Information, Dalian 116023 China; jianing@neusoft.edu.cn
\*   Correspondence: zhengchunjun@neusoft.edu.cn

**Abstract:** Speech emotion recognition is a challenging and widely examined research topic in the field of speech processing. The accuracy of existing models in speech emotion recognition tasks is not high, and the generalization ability is not strong. Since the feature set and model design of effective speech directly affect the accuracy of speech emotion recognition, research on features and models is important. Because emotional expression is often correlated with the global features, local features, and model design of speech, it is often difficult to find a universal solution for effective speech emotion recognition. Based on this, the main research purpose of this paper is to generate general emotion features in speech signals from different angles, and use the ensemble learning model to perform emotion recognition tasks. It is divided into the following aspects: (1) Three expert roles of speech emotion recognition are designed. Expert 1 focuses on three-dimensional feature extraction of local signals; expert 2 focuses on extraction of comprehensive information in local data; and expert 3 emphasizes global features: acoustic feature descriptors (low-level descriptors (LLDs)), high-level statistics functionals (HSFs), and local features and their timing relationships. A single-/multiple-level deep learning model that meets expert characteristics is designed for each expert, including convolutional neural network (CNN), bi-directional long short-term memory (BLSTM), and gated recurrent unit (GRU). Convolutional recurrent neural network (CRNN), based on a combination of an attention mechanism, is used for internal training of experts. (2) By designing an ensemble learning model, each expert can play to its own advantages and evaluate speech emotions from different focuses. (3) Through experiments, the performance of various experts and ensemble learning models in emotion recognition is compared in the Interactive Emotional Dyadic Motion Capture (IEMOCAP) corpus and the validity of the proposed model is verified.

**Keywords:** ensemble learning; multi-level technology; acoustic features; speech emotion recognition; deep learning model

## 1. Introduction

As the most convenient and natural medium for human communication, speech is the most basic and direct way we have to transmit information to each other. Speech contains a variety of information, and can express rich emotional information through the emotions that it carries and its perception in response to objects, scenes, or events. As an important research direction for speech signals, Speech Emotion Recognition (SER) is key for computers to understand human emotions and is the premise of human–computer interactions [1].

Speech emotion recognition aims to identify the correct emotional state of the speaker through the speech signal. At present, research on emotion is still an interdisciplinary field, and it has not yet

been uniformly defined and standardized. Since speech is not a complete expression of emotional physiological signals, how to efficiently and accurately recognize the emotions expressed by people while ignoring other sensory information is a hotspot in the field of phonetics research in recent years.

At present, the recognition rate of speech emotions is low, and the generalization ability is not strong. It mainly comes from the following aspects:

First of all, the features involved in existing emotion recognition are mainly global features [2]. Even if corresponding statistical functions are added, there is still less consideration of the temporal characteristics of speech. Existing research is based on specific languages, and emotion recognition based on this idea is not universal. The decision-making aspect of the speech emotion recognition model often plays a decisive role. At this time, if the state of the expert is unstable, it directly affects the final emotional judgment. The speech emotion recognition model often involves the traditional pattern recognition method. Since there is no universal speech emotion model at present, the accuracy of the model is greatly affected. In addition, the corpus available for testing is limited in size and number, making it difficult to find a fully matched comprehensive training model.

Based on the above research status, although some scholars are working to overcome these problems to improve the recognition rate of speech emotions, few experts have fully explored the correlation between global and local features in different roles, features, and models. Focusing on the above problems, this paper carries out related research on the design of speech emotion features with a multi-level deep learning model and constructed ensemble learning schemes for the comprehensive consideration of multi experts' suggestions [3]. A more stable ensemble learning model with comprehensive generalization ability and evaluation of speech emotions is proposed.

The second part of the paper focuses on a feasibility analysis of popular acoustic feature extraction methods and related deep learning models. In the third part, three experts in speech emotion recognition are designed according to different angles describing the acoustic characteristics. Each expert, also known as a classifier, separately extracts multidimensional features and designs a single-/multiple-level deep learning model that meets its characteristics, and then trains them separately. At the same time, an ensemble learning program is designed to enable the three experts to evaluate speech emotion from different sides. The fourth part lists the results of model training and testing of the three experts. Based on a comparison of their respective performances, the ensemble learning test is conducted. The fifth part is the summary of the work of this paper and the prospects for future work.

## 2. Related Work

In the field of speech emotion recognition, it is known that many features can express the emotion of speech. If the unique advantages of different speech emotion models are combined and the features are fused together, the recognition performance can be effectively improved. However, in practice, a simple series/parallel connection of two speech emotion features is generally adopted. This processing method leads directly to increased dimensions of the speech emotional feature after fusion, which excessively increases the calculation amount of the entire recognition process, thereby invisibly increasing the spatial and time complexity of the recognition system [4].

The deep learning method can learn nonlinear representations of effective speech signals from different levels of input. It has been widely used in voiceprint recognition, speech recognition, and emotion recognition. Currently, there are two types of deep learning models, supervised and unsupervised. In the face of speech emotion recognition tasks, deep neural network (DNN), convolutional neural network (CNN), recurrent neural network (RNN), and other supervised models are often used. To highlight the signal characteristics of different tasks, multi-level recognition technology and an attention mechanism are also integrated to perform emotion recognition.

### 2.1. Voiceprint Recognition Technology

In foreign countries, the DNN model is widely used in the study of emotion recognition. The DNN model uses personalized features as input [5]. In general, personalized features carry a large amount of

personal emotional information, reflect the characteristics of the speaker, and do not contain common emotional information, content, and environment of different speakers [6]. The performance of the time delay neural network (TDNN)-statistics pooling, TDNN long short-term memory (LSTM), TDNN-LSTM-attention, LSTM, and LSTM-attention models were compared in [7]. It was found that TDNN-LSTM based on the attention mechanism obtained the best emotional recognition results. Researchers abroad [8] pointed out that the CNN model can also be applied to low-level acoustic features to identify areas of emotional significance. Few people have defined or applied the statistical functions of statements because of the confusing information, and Mel filter banks are generally used as a substitute. In [9], a convolution-pooling network was proposed to identify the emotionally significant regions in variable-length speech, and the extracted features have higher emotion recognition accuracy. In view of the problem that it is easy for the pooling layer of CNN to lose information, Aldeneh et al. [10] found that the maximum pooling layer can be used because it has the best effect on emotion recognition and has the characteristic of translation invariance.

Due to the limitations of the CNN model, the fusion features it obtains are often limited to spectrograms or low-level descriptors (LLDs). These features ignore the important characteristics of the speech signal; they are collections of units with time series. As the most popular new architecture in acoustic missions [11], RNN models are often used in conjunction with spectral and prosodic features. At the same time, RNN adds a self-joining form related to time nodes, highlighting its ability to model time series. Lee et al. [12] applied the RNN model to learn temporal and spatial relationships of emotion recognition. An attempt was made in [13] to add an attention mechanism to the LSTM layer to find important time periods of speech signals associated with emotion recognition.

At present, at home, most of the research on SER is based on personal emotional characteristics, and has shown good recognition performance, but the practical application of SER technology in real environments that are independent of the speaker is hindered. Therefore, reducing the differences in speech style characteristics is important for speech emotion research. Mao et al. [6] used CNN to learn support vector machine in order to find salient features of classification. Due to the tendency of overfitting during training, Huang et al. [14] proposed shake-shake regularized residual network (ResNet) for speech emotion recognition. Through a combination of various streams of data and different parts of the feature space to combat noise, it was proved that it could improve Unweighted Accuracy (UA) and reduce overfitting, which is beneficial for learning in emotional computing.

When manually extracting features for the CNN model, the problem of correlation between signals is often ignored. An effective emotion recognition system based on deep convolutional neural network (DCNN) is proposed in [15]. The logarithmic spectrum is calculated and principal component analysis (PCA) is used to reduce dimensionality and suppress interference, and the emotional information is learned from the labeled training speech segments. In [16], a deep retinal convolutional neural network is proposed for SER, with advanced features learned from a spectrogram, which is superior to previous studies on the accuracy of emotion recognition.

Han et al. [17] used a variety of methods to solve the problem of LSTM output length and label length mismatch, and adopted the connectionist temporal classification (CTC) mechanism to improve learning efficiency. At present, common RNN-related models include bi-directional RNN, convolutional recurrent neural network (CRNN), multidimensional cyclic neural network, LSTM, and gated recurrent unit (GRU). They all focus on dealing with complex signals of the past and future in different periods. Therefore, their inputs are often related to the original signal characteristics, which is quite different from CNN's method of processing spectral features.

## 2.2. Multi-Level Recognition Technology

Although both CNN and RNN are supervised models, they have their own characteristics and areas of expertise, and the characteristics of the inputs are not the same. In recent years, more researchers have effectively combined the models, trying to exploit the characteristics of all models and improve the accuracy of emotion recognition. In foreign countries, Keren et al. [18] combined CNN with

LSTM to improve speech emotion recognition based on the Mel filter or original signal. Ma et al. [19] applied CNN to the spectrogram of variable-length speech segments, and combined CNN with RNN to complete the emotion recognition task, which improved the accuracy of speech segmentation.

At the same time, in China, Luo et al. [20] studied two joint representation learning structures based on CRNN, aiming to capture emotional information from speech. A two-level SER system was obtained by combining various forms, and joint learning was performed by using spectral pictures of time scales.

Recently, the attention mechanism has also begun to be integrated into relevant models. On the basis of improving efficiency, the ability to explore distinctive regions is increased, so that efficient identification can be achieved with the most effective features and the shortest time. The point of view that attention can help to quickly analyze information and focus on specific information about the data is proposed in [21]. In [22], a three-dimensional attention-based convolution recurrent neural network is proposed to learn the discriminative features of SER, which not only retains effective emotional information, but also reduces the influence of emotional independence.

### 2.3. Ensemble Learning Technology

Through the fusion of multiple levels, the advantages of different neural networks are connected in series, so as to achieve the goal of improving recognition efficiency from different angles. However, if the trained model is applied to the test platform, it is easy to cause the phenomena of gradient disappearance and overfitting. Therefore, improving generalization ability is important in speech emotion recognition. Ensemble learning has strong generalization ability and parallelism, which is conducive to improving the reliability and efficiency of identification. The accuracy and credibility of each expert are important in the voting strategy of ensemble learning.

In recent years, experts have utilized ensemble learning with traditional machine learning methods in the field of speech emotion recognition. Mao et al. [23] separated rich linguistic features from emotional features in speech, then used adaptive weight fusion to prove that rich linguistic features are conducive to improving the recognition rate. Liu et al. [24] used different types of emotional feature subsets to train subclassifiers, and used the subclassifiers to achieve decision-making layer fusion, obtaining better recognition results. However, the existing ensemble learning is mainly based on the distribution of expert credibility. However, for expert decision-making, the root of the data is the characteristics of speech, the form of acquisition is single, and subtle differences between samples are likely to lead to inaccurate clustering information.

Based on this, multiple forms of models can be used to provide more stable multi-angle, multi-expert decisions for ensemble learning models. At the same time, according to the accuracy rate of each expert, the credibility is adjusted online. The generalization ability is improved along with the speech emotion recognition.

## 3. Proposed Method

### 3.1. Design Route for the Overall Model

The overall model framework adopts the model structure of ensemble learning, which involves 3 emotional judgment experts with different focuses, which make emotional decisions in their respective fields. Through ensemble learning, the emotional decision with the highest recognition rate is finally obtained.

As an important branch of information fusion, ensemble learning is generally used for classifier integration. Its goal is to integrate multiple kinds of information to obtain better classification accuracy. According to the characteristics of the differences in emotional speech, the fusion method of weighted voting at the decision level can be adopted; that is, the output of multiple experts is negotiated according to the customary rules, and finally a unified opinion is reached. The emotional marker with the highest number of votes is used as the final result of the sample to be tested.

Assuming that the sample to be tested is $x$ and the number of votes identified as $W_j$ is $v_j$, then the formula for weighted voting is as shown in Equation (1):

$$v_j(x) = \sum_{j=1}^{m} W_j d_j(x) \qquad (1)$$

Here, the weight is designed in the voting process and is determined according to the training data to estimate the confidence of the classifier. $d_j(x)$ indicates that sample $x$ is determined by the NO. $j$ expert as a weighted label of the emotion category $W_j$.

If the accuracy of the voting results of each expert is improved, the accuracy of the recognition can be improved according to this design idea. Once some experts have scored mistakes, their confidence would be reduced, and other experts lead the recognition of speech emotions. This design idea has certain generalization capability. In summary, in the above scheme, the accuracy and confidence distribution method of the expert is the key element, and it is also the content and experimental point to be described later.

Expert 1: A spectrogram is a kind of 3-dimensional spectrum that contains rich time and frequency information. The expert enhances the data of the original speech signal. The data source of the model is the map of the original audio generated under various amplitudes, the time domain, and the frequency domain transformed map. Then they are fed into a 2-channel CNN model. Expert 1 can make more effective use of the time-domain and frequency-domain features of speech to highlight the expression of local features.

Expert 2: Based on the characteristics of short-term invariance of speech, expert 2 focuses on the subdivided audio files. After processing the PCA, the mean value features of the spectrogram can be fused. Speech is sequential, so it is suitable for GRU, which is a deep model of time series. Combined with the attention mechanism, the model can further enhance the expression of global features of short-term speech.

Expert 3: LLD features and their HSFs can describe features from a global point of view, and can be used as a powerful supplement to existing time series features. Therefore, for segmented speech, the local original speech signal is fed into the CRNN model based on the features of the speech spectrum, generating an abstract representation of advanced features obtained from the deep learning model, which is then combined with HSFs, and the joint representation is fed into the BLSTM model, which is suitable for temporal data. This expert extracts global features of speech from multiple perspectives.

Each expert's focus and input are different. When setting the confidence level, comprehensive consideration is needed. The design method of confidence is described in detail later.

Considering that the recognition performance of the deep neural network requires high data volume, a large amount of new data is generated in the preprocessing segment, which can minimize the impact on the model due to insufficient data volume, and can also speed up the network training and reduce the impact of overfitting during training on small datasets.

### 3.2. Expert 1: Double-Channel Model Based on CNN

Expert 1 focuses on local feature information, using the original speech information as input and converting that information into the corresponding spectrogram. To ensure diversity of the input signal, the current spectrogram incorporates multiple forms of speech information. Since the spectrogram obtained by the same speech information has a strong correlation, a double-channel model in the form of a parallel CNN is selected to process the results of the speech information without using a tandem form.

The features extracted from the spectrogram can be used to recognize the arousal of emotional speech. In the classification of the arousal, the classifications of happy and angry are higher, while those of neutral and sad are lower. Based on this, expert 1 as designed in this paper takes happy and

angry as one category and neutral and sad as another. Expert 1 is only used to complete the task of two categories.

### 3.2.1. Analysis and Preprocessing of Speech Signals

Analysis and preprocessing of speech signals are two important methods. The analysis of time and frequency domains has its own characteristics, but it also has certain limitations. Time domain analysis does not have an intuitive understanding of signal frequency characteristics. Frequency domain characteristics do not reflect the relationship of speech over time.

A spectrogram combines the characteristics of the spectrum and the time domain waveform to dynamically display changes in the speech spectrum over time. Therefore, more information is included in the spectrogram than in the simple time or frequency domain. The vertical axis of the spectrogram represents frequency, the horizontal axis represents time, and the depth of the color represents the energy of the signal of the arbitrary specific frequency component at the corresponding time.

Although, as a whole, the characterization and essential characteristic parameters of speech signals change over time, this seems to be a nonstationary process. However, in this research, it is found that the characteristics of speech signals in a short time can be regarded as remaining unchanged, that is, they have a smoothness. Therefore, when they are preprocessed, short-time stationary technology is generally adopted. Short-term analysis is carried out to eliminate the redundant part of the signal, and it is split into units of segments to analyze its characteristic parameters. Therefore, before acquiring the spectrogram, the existing speech in the Interactive Emotional Dyadic Motion Capture (IEMOCAP) corpus is first segmented in the time domain and combined with the characteristics of the corpus speech, the speech is segmented into units of 1 s, and then the spectrogram is stretched or compressed in the time domain to achieve data expansion and generate many spectral maps under various amplitudes. At the same time, based on the rotational invariance of the CNN model, the time and frequency domain information of the spectrogram are flipped from multiple angles, in order to add a new set of basic data after transformation. The goal of this design method is to extract time-frequency features and highlight the expression of local features.

Through the above transformation, 2 sets of homologous spectrograms are obtained. To feed them into the CNN model, the spectrograms need to be adjusted to a uniform size.

### 3.2.2. Design of Double-Channel Model Based on CNN

A convolutional neural network is a deep neural network in which alternating convolution and pooled layers are stacked. The role of the convolution layer is to extract features. The neural unit of the current layer is connected to several feature maps of the previous layer through a set of convolution kernels, the convolution operation is performed, and the feature map of the current layer is obtained by adding the bias.

The convolution kernel is shared by all neural units of the same feature map, which is called weight sharing, greatly reducing the size of the parameters. Each neural unit is only connected to a local area of the previous feature map. Each neural unit extracts features of the local area, and all of the neural units are combined to obtain global features.

To obtain more comprehensive information from the feature parameters, multiple convolution kernels are used in the same layer network to obtain multiple feature maps.

After the convolution operation, the feature map performs a down sampling operation at the pooling layer. The pooling unit calculates the main information of the local area in the feature map, removes the redundant information, and reduces the scale of the operation.

Based on the above CNN model structure, a double-channel model structure was designed, as shown in Figure 1. Each channel of this model is composed of a CNN model, in which the input data source is different, which is the spectrogram generated under various amplitudes, the time and frequency domain transformed image. Since the CNN structure requires large-scale training data,

it is necessary to dynamically generate different forms of pictures for the time and frequency domain features of the spectrogram.

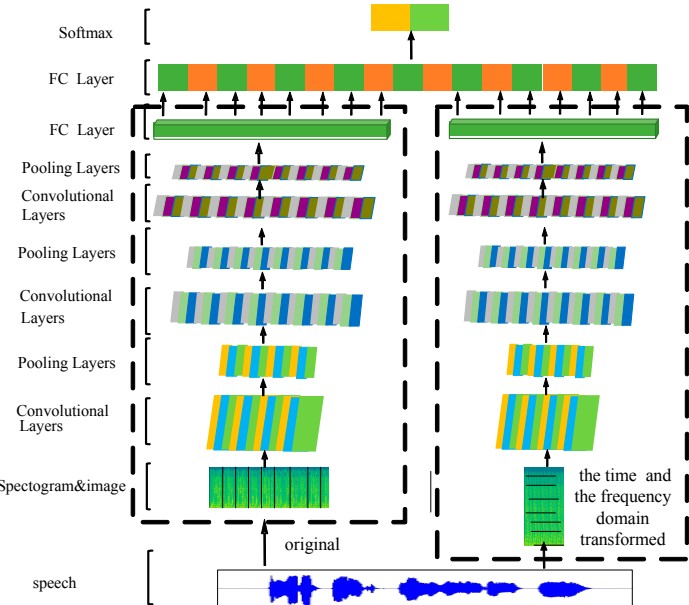

**Figure 1.** Double-channel model structure based on convolutional neural network (CNN).

Each channel has a similar CNN structure except for the difference in the processing scheme of the spectrum. The CNN structure consists of 3 convolutional layers, 3 pooled layers, and 2 fully connected layers, for a total of 8 layers. The input information size of the first convolutional layer is $310 \times 310 \times 3$, where 310 is the length and width of the spectrum and 3 is the 3 channels red (R), green (G), and blue (B). The spectrogram passes through 64 convolution kernels with a size of $3 \times 3$. After the convolution operation with a step size of 1, 64 feature maps are generated, and then the rectified linear unit (ReLu) activate function is used. After the maximum pooling operation, 64 feature maps are obtained. The input source of the second convolutional layer is the output characteristic values of the first layer. The calculation process is the same as the first layer. The third layer is the same, followed by a fully connected layer, which has a total of 1024 neurons. Finally, a dropout operation is performed on this layer to prevent overfitting of the model.

After the first 2 layers are fully connected in the above 2 channels, all the outputs are combined as the input of the next fully connected layer, and the emotional class value is obtained by the softmax function.

Through the combination of multiple levels of information, the parameters learned by the network cannot change drastically in the local area, and the generalization ability of the model for data feature learning is improved.

### 3.3. Expert 2: GRU Model Combining the Attention Mechanism

Unlike expert 1, expert 2 focuses on extracting comprehensive features within the window, involving the acquisition and processing of information such as the mean of the data within the window. In the feature extraction of expert 2, a variety of processing methods are adopted. Then, the features are fed into the GRU model to form an expert with a high recognition accuracy.

3.3.1. Local Comprehensive Feature Extraction

Expert 2 also uses the original speech signal as input, and converts the speech information into corresponding spectrogram data on the basis of speech segmentation. To maintain the original characteristics of the input signal to the greatest extent, only the original spectral data are selected,

the time and frequency domain information are retained, and the depth information in the map is discarded, and the PCA operation is performed.

PCA can map high-dimensional data into low-dimensional space, perform feature decomposition work through covariance matrix, obtain the main components and weights of the data, and select the features with large k-dimensional variance, and the projected data can satisfy the characteristics of maximal variance. The distribution of data points at this time is sparse, which is convenient for extracting core data. This method of data dimensionality reduction can maximize the proximity to the original data, but it does not focus on exploring the internal structural characteristics of the data.

Special local processing work is performed on the generated features. The width of the generated spectrum is 513 dimensions. As the horizontal axis unit, the size of each set of feature files is calculated to be $32 \times 513$ dimensions. The features are longitudinally combined to calculate the mean of each column feature to form a global feature within the $1 \times 513$-dimension window. The result of this processing method reflects the comprehensive information inside the current window. In this way, the important data inside the window are extracted and the effect of the mute area on the current window is eliminated.

### 3.3.2. Design of GRU Model Combined with the Attention Mechanism

GRU is an enhanced version of the LSTM network, which is simpler and has a very streamlined structure. As a variant of LSTM, GRU can also solve the problem of long dependence in RNN networks. Unlike LSTM, there are 2 types of gates in the GRU model: update gate and reset gate.

The GRU model combines the forgotten gate and input gate into a single update gate, and merges the cell state with the hidden state. The update gate is used to control the extent to which the status information of the previous moment is brought into the current state. The amount of information that the previous state is written to the current candidate set is controlled by the reset gate. The update gate also mixes the cell state and hidden state, and simplifies the standard version of the LSTM model through formal changes.

Considering that the energy of speech is a process of concentrated explosion in a short time, it is possible to focus on the key areas of the outbreak for feature selection and training, and to increase the contribution of such areas. Therefore, on the basis of the GRU model, the attention mechanism is added, which is a selection mechanism for assigning limited information processing capabilities that helps to quickly analyze target data and cooperates with information screening and weight setting mechanisms to improve the computing power of the model.

For each vector $x_i$ in the sequence of input $x$, attention weight $\alpha_i$ can be calculated according to Equation (2), where $f(x_i)$ is the scoring function:

$$\alpha_i = \frac{\exp(f(x_i))}{\sum_j \exp\left(f(x_j)\right)} \tag{2}$$

The output of the attention layer, *attentive_x*, is the sum of the weights of the input sequences, as shown in Equation (3):

$$attentive\_x = \sum_i \alpha_i x_i \tag{3}$$

The second expert chooses to introduce the attention mechanism after the 2-layer bi-directional GRU model. The specific model structure is shown in Figure 2.

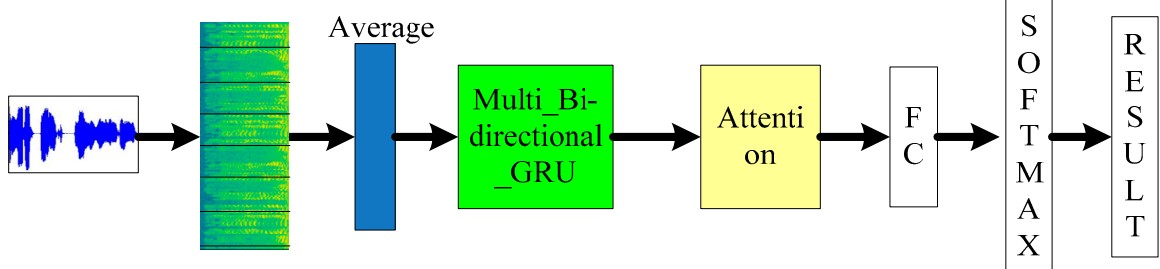

**Figure 2.** Gated recurrent unit (GRU) model combined with the attention mechanism.

As shown in Figure 2, the spectrum map is first processed in order to reduce the dimension, merge vertically, and average the PCA. After experiencing the 3-layer bi-directional GRU model, the attention mechanism is fed to obtain the salient features. Then, the full connection layer is experienced, and the judgment result of each emotion is obtained as the discrimination result of the second expert.

### 3.4. Expert 3: Multi-Level Model Based on HSFs and CRNN

On the basis of the first 2 experts, expert 3 carries out feature fusion of various levels. In addition to pre-fusion of the spectrogram and the model, LLD and its related HSFs are also introduced. Finally, the 2 types of features are used together to train the model.

#### 3.4.1. Feature Selection and Integration

In the task of speech emotion recognition, the original speech features commonly used include Mel-frequency cepstral coefficient (MFCC), fundamental frequency, energy characteristics, and formants. The existing methods of fusing speech emotion features often have problems such as increased space and time complexity. To effectively solve the problem of excessive dimensionality, the dimension features with the greatest contribution to speech emotion recognition can be selected from the speech emotion feature parameters.

Table 1 lists the most important 20-dimensional original speech features and their commonly used HSF information. Through the feature selection method, it is determined that the features involved in Table 1 make the greatest contribution in the process of speech emotion recognition.

**Table 1.** Set of low-level descriptor (LLD) features.

| | Features |
|---|---|
| LLD | Loudness, spectral flux, MFCC1-2, alpha ratio UV, equivalent sound level, F0 semitone, F1 frequency, slope V0–500, shimmer, local dB, logMelFreqBand1–6 |
| HSF | sma, mean, stddev |

Spectral features are recognized as characteristic parameters based on the auditory characteristics of the human ear and the mechanism of speech production. The extraction principle of the MFCC is designed based on the ear's auditory mechanism. It is the most frequently used and most effective spectral feature in speech emotion recognition.

The MFCC can be composed of 13 parameters, delta and delta delta. In the basic discriminative environment of sentiment classification, the validity of the first and second parameters of MFCC is the highest.

In addition to the MFCC, the Alpha Ratio is obtained by using energy of 50–1000 Hz and 1–5 kHz and the Hammarberg index, obtained by dividing the strongest energy peak of 0–2 kHz with the strongest energy peak of 2–5 kHz. Spectral slope 0–500 Hz, the logarithmic power of the Mel band, and spectral flux are also extracted as important parts of the feature set.

Since the pitch frequency has a certain relationship with the individual's physiological structure, age, gender, and pronunciation habits, it can generally be used to mark the person's emotional expression.

The pitch period is the length of time that the vocal cords vibrate during pronunciation. When a voiced sound is produced, a quasi-periodic excitation pulse train is generated, resulting in vibration of the vocal cords.

Energy characteristics are generally related to sound quality. They describe the nature of the glottis excitation signal, including the vocalist's voice and breathing; the performance of energy characteristics varies from emotion to emotion. By evaluating them, the emotional state can be distinguished. In addition, Shimmer and Loudness energy features are selected. Shimmer represents the difference in amplitude peaks between adjacent pitch periods. An estimate of Loudness that can be obtained from the spectrum, the spectral flux of two adjacent frames, can be calculated from the energy.

Equivalent Sound Level is a way to describe the level of sound over time. It can track all fluctuations, calculate the average energy at the end of the measurement, obtain the information in decibels, and then take the logarithm. At the level of valence, this feature is more accurate for describing speech with greater mood swings.

Performing an HSF representation on the LLD features to obtain global features of the segmented speech information, the HSF representation methods used here are sma, sma mean, and stddev. Sma represents a smooth global result by a moving average filter with window length n. Mean represents the mean of LLD, and stddev represents the standard deviation of LLD. In addition, statistical functions such as rising slope and 20th percentile are used.

For global features, the LLD and its HSF results are extracted in batches to obtain the best combination and a simplified version of the feature set, which is the first level input of expert 3.

### 3.4.2. Design of Feature Extraction Model Based on CRNN

When modeling speech signals, time-dependent models are often used, highlighting the effects of significant regions. Aiming at the requirements, a speech emotion recognition model based on CRNN is proposed. The model is mainly designed for the time domain features of the spectrogram. On the basis of segmentation of speech, the mode of spatial spectrum representing emotional information was effectively studied, and a multi-layer bi-directional LSTM structure was designed. The model is shown in Figure 3. The output of this model was used as the input to the second level of expert 3, which represents local features with timing relationships.

The CRNN model consists of 2 parts, CNN and RNN. The former is based on the traditional CNN model, which is mainly used to extract the frequency domain features of the spectrogram. For pre-segmented speech, the features of CNN learning for each segment can be obtained. The input image is convoluted into 6 steps (3 consecutive sets of convolution and pooling operations) to generate a set of features. Then the features are sent to the RNN model, where a multi-layer bi-directional LSTM network is used, where each time step corresponds to a segment of the original speech input, thereby preserving long-term dependencies between regions.

The goal of designing the CRNN model is not to directly judge the speech emotion, but to use the model obtained from the CRNN training for secondary verification. When verifying, the result is directly computed after the penultimate full connection. The result is used as the local feature for subsequent feature fusion. The model takes into account the characteristics of the time and frequency domains of the spectrogram. Using the spectrogram as the input of the network, the parameters of the CNN convolutional layer are trained, and the feature map output by the CNN is reconstructed. At this point, the advantages of CNN for image recognition and the characteristics of the RNN's ability to process serialized data are fully utilized.

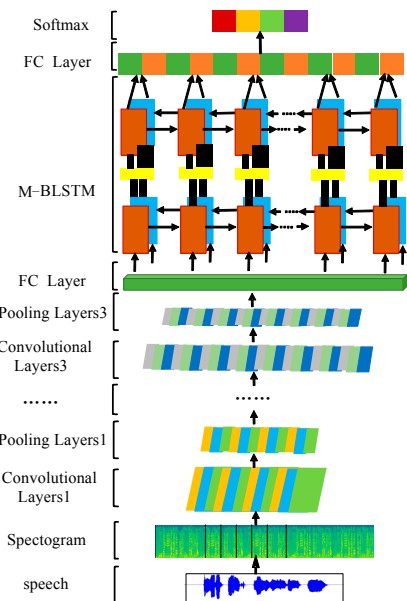

**Figure 3.** Convolutional recurrent neural network (CRNN) model.

### 3.4.3. Design of Multilevel Model Based on HSFs and CRNN

Features learned through the LLD-HSF and CRNN models describe the state of emotions from different aspects, and there are complementary characteristics between them. Combining the above learned features, expert 3 was designed and a multilevel emotion recognition model was proposed. Two types of features are connected through the hidden layer and are projected into the same space. Compared with the traditional way, the discrimination of emotion-related features is enhanced.

As shown in Figure 4, the entire multi-level model is trained in an end-to-end manner, with given segmented speech being processed simultaneously on 2 parallel levels. On one level, these segments are input to the CRNN based on the spectrogram segmentation. The output of the CRNN model is a $32 \times 513$-dimensional vector, which is then added to an attention layer and a hidden layer of 128 dimensions, and the high-dimensional features are successively mapped to the low-dimensional 1024-dimensional feature space. On another level, the original waveform is segmented into frames. The LLD feature is extracted from each frame, and the corresponding HSFs are further counted to obtain a global feature vector with a size of 20 dimensions.

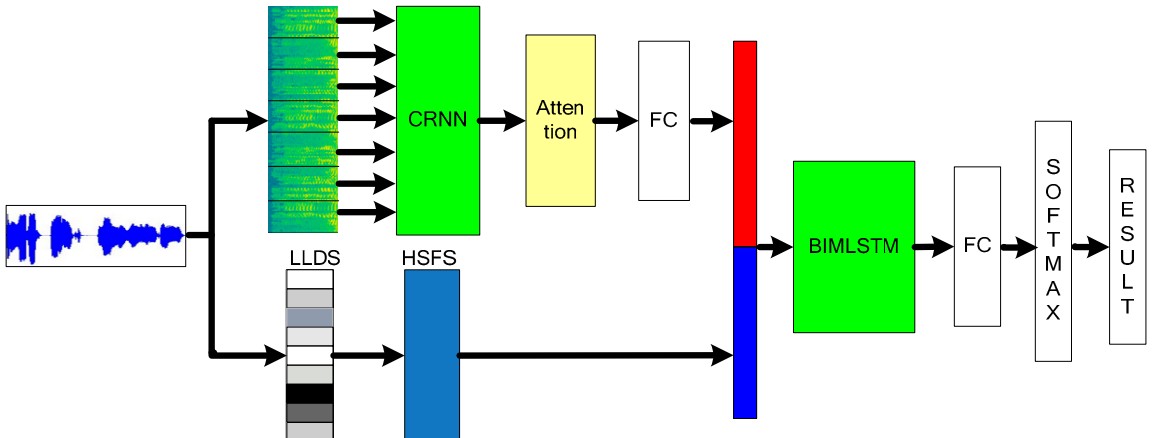

**Figure 4.** Joint model of high-level statistics functions (HSFs) and CRNN.

By linking the feature vectors of the 2 levels, on the basis of integrating the 2 types of features, they are sequentially added to a bi-directional 3-layer LSTM model, an attention channel, and a hidden

layer. The hidden layer projects them into the feature space, passes them to the Softmax layer for classification, and finally outputs the classification results.

Expert 3 is characterized by the effective combination of local and global features. Local features can maximize the weight of features of certain important areas, while global features measure the standard of speech over the entire time range. Both types of features have their own characteristics and focus on the direction. An effective combination can increase the significant area while taking into account the information of all features. Multi-level fusion can improve the basic elements of local and global features. On the basis of combining attention mechanisms, the salient regions in the input features are reversely acquired, and the recognition accuracy is improved in the reverse direction.

### 3.5. Design of Ensemble Learning Model

For the speech emotion recognition task, 3 experts were designed from different angles. Each expert has a different emphasis. Designing an ensemble learning model is equivalent to completing an emotional vote by multiple experts.

Using the existing training modules and 5-fold cross-validation, each time the training data are randomly divided into 5 groups, and one group of data is extracted for testing and the remaining samples are used for training; a total of m times are extracted, and a plurality of sampling sets for training are obtained. Since the verification result for the training set can be obtained for each training, the values of WA and UA can be obtained each time, and the following formula is obtained:

$$UA_j = \frac{1}{n}\sum_{i=0}^{\exp}\sum_{j=0}^{m} UA_{ii} \tag{4}$$

$$UA\_\exp_i = \frac{1}{\exp i}\sum_{j=0}^{i} UA_j \tag{5}$$

$$1 = \sum_{i=i}^{\exp} W_{\exp i} UA\_\exp_i \tag{6}$$

where $\exp_i$ ($i$ = 1, 2, exp) is the number of experts in each category, exp is the number of expert categories, $n$ is the number of emotion classes, $UA_{ii}$ is the UA of each expert under each category, $UA_j$ represents the average UA of each expert, $UA\_\exp_i$ is the average UA of each type of expert, and $W_{\exp i}$ is calculated by normalizing the UA, which represents the confidence of each category of experts.

For 3 experts to re-rank, $W_{\exp i}$ is sorted by numerical value, $W_{\exp 1}$ has the highest accuracy, and $W_{\exp 3}$ has the lowest accuracy. Since the spectrogram of expert 1 is susceptible to the surrounding environment and noise during application, its effect is usually the worst. To improve the overall efficiency, it is necessary to carry out multiple binary classifications. $W_{\exp 31}$ and $W_{\exp 32}$ represent the results of the second binary classification of the expert where $W_{\exp 3}$ belonged.

The overall ensemble learning program is as follows:

1. In the 4-category task, if the opinions of the experts to which $W_{\exp 1}$ and $W_{\exp 2}$ belong are the same, the result of the expert to which $W_{\exp 1}$ belongs is the final result.
2. In the 2-category task, the opinions of the experts to which $W_{\exp 1}$ and $W_{\exp 2}$ belong are consistent. In the 4-category task, the expert opinions of $W_{\exp 1}$ and $W_{\exp 2}$ belong to the opposite, and the classification of the value of $\max(W_{\exp 1i} + W_{\exp 3i}, W_{\exp 2j} + W_{\exp 3j})$ is the final result, where $i$ and $j$ represent the views of the experts to which $W_{\exp 1}$ and $W_{\exp 2}$ belong, respectively.
3. In the 2-category task, the opinions of the experts to which $W_{\exp 1}$ and $W_{\exp 2}$ belong are inconsistent. At this point, it is necessary to observe the expert opinion of $W_{\exp 3}$, judge the first 2 experts to find the expert with the same result as the first classification of the $W_{\exp 3}$ expert, and use its result as the final result.

By traversing the results of each expert for each piece of data, the above-mentioned schemes are used to judge, and the results predicted by the integration experts are obtained.

For the more balanced samples of each type of collection, the above method can quickly obtain the ensemble learning results of the experts. If the sample data distribution is skewed or the recognition rate of a certain category is too high or too low, there will be some errors in the results obtained. At this point, the emotional classification of the expert to which $W_{\exp 3}$ belongs can be further refined, by performing the second classification to eliminate the negative impact of certain emotional accuracy when it is not credible. The above is the first scheme of the ensemble learning.

At this point, step 2 of the above scheme is refined into the following process: If the views of experts to which $W_{\exp 1}$ and $W_{\exp 2}$ belong to are the same in the 2-category task, and the views are inconsistent in the 4-category task, first judge the emotional classification of the expert to which $W_{\exp 3}$ belongs; if it is the same as the 2 classifications of the other 2 experts, the classification with the value of $\max(W_{\exp 1i} + W_{\exp 3i}, W_{\exp 2j} + W_{\exp 3j})$ is the final result. On the contrary, the expert to which $W_{\exp 1}$ belongs is the final result. In this way, the second scheme of the ensemble learning is formed.

## 4. Design of Experiments and Analysis of Results

### 4.1. Experimental Preparation

The IEMOCAP dataset, which was designed to study the interaction of binary expressions in multiple modes, was used for the experiments. It was collected in 5 binary sessions of 10 themes using motion capture and audio/video recording. Each session consists of a different conversation in which a male and a female actor execute a script and participate in a spontaneous impromptu conversation triggered by an emotional scene cue. At least three evaluators used the classified emotional tags selected from the collection to annotate each utterance in the dataset. The optional categories were happy, sad, neutral, angry, astonished, excited, frustrated, disgusted, fearful, and others. When at least two-thirds of the evaluators gave the same emotional tag, the existing types of scripts and impromptu conversations in the dataset were no longer subdivided.

For the 10,039 items of standard voice data in this dataset, related similar emotions were combined to remove less correlated samples. Finally, four types of emotional sample data are summarized. The excited and happy categories were combined to form a new happy category. In addition, there are sad categories, angry categories, and neutral categories. The remaining 6 categories of sample data were discarded. Based on the classification method, a total of 5531 samples were retained, with the following number belonging to each type: angry: 1103; happy: 1636; neutral: 1708; and sad: 1084.

Then, in units of 1 s, the batch of samples was subjected to basic segmentation, and a total of 21,517 pieces of sample data was obtained. After the division, the numbers of samples were: angry: 3448; happy: 5181; neutral: 4313; and sad: 4270. Aside from the small size of the angry category, the data volume of the other categories was more balanced. In the preprocessing of this corpus, the speech segmentation length used was less than the traditional average segmentation length of 2.5 s, based on the sample data volume balance of this corpus. Experiments were performed using five-fold cross-validation; 80% of the data was used to train deep neural networks, and the remaining 20% was used for testing and verification of accuracy.

When designing the overall model, each spectrogram has a size of $32 \times 513$ dimensions. In the CNN and CRNN models, the batch size is 100 and the maximum number of training items is 100,000. The learning rate is set to 0.001 and the dropout is set to 0.7. In addition, Adam is used as the optimizer. The mean square error is used as a loss function.

The CNN infrastructure consists of three convolutional layers. The first layer has 64 filters of size $3 \times 3$, and layers 2 and 3 contain 96 filters of size $5 \times 5$. The ReLu function is used for all three convolutional layers. The size of the two largest pooling layers is $2 \times 2$, and the size of the stride is 1. There are 2 nodes in the Softmax layer, and cross entropy is used as the objective function for training.

In the LSTM model, a bi-directional three-layer structure is adopted. The batch size is 300. The maximum number of training items, learning rate, loss function, and optimizer are the same as CNN, and the dropout is set to 0.5. In the HSF level, 16 LLD features and 4 HSFs are selected to form a 20-dimensional feature set.

## 4.2. Independent Experiment for Each Expert

Many experimental verifications were carried out for each expert. The TensorFlow framework was used to build the network model structure. Experiments were performed on the spectrogram features and low-level descriptor features. To avoid the impact of different emotional imbalances, weighted accuracy (WA) and unweighted accuracy (UA) were used as indicators. The different sentiment classification models were tested. The first experiments focused on the features of the spectrogram, comparing the effectiveness of each expert.

Experiment 1: Mainly used for testing expert 1, with 21,517 samples + spectrogram + CNN model as the baseline; the feature information does not use any processing method.

Model 1: 5531 samples + spectrogram + CNN model; model 2: 2 × 21,517 samples + spectrogram + CNN model (two-layer convolution and two-layer pooling); the original picture adds time and frequency domain transformation. Model 3: 3 × 21,517 samples + spectrogram + CNN model; feature information adds original picture and PCA channel. Model 4: Expert 1, 8 × 21,517 samples + spectrogram + CNN model, combined with time and frequency domain transformation. The accuracy of each model's emotion recognition was calculated separately. Table 2 shows the accuracy of the speech emotion recognition models after experimental verification.

**Table 2.** Test results of expert 1.

| Model/Sample Set | WA | UA |
|---|---|---|
| Baseline: 21,517 samples | 46% | 43% |
| Model 1: 5531 samples | 33% | 34% |
| Model 2: 2 × 21,517 samples | 54% | 53% |
| Model 3: 3 × 21,517 samples | 50% | 51% |
| Current System: Expert 1, 8 × 21,517 samples | 69% | 68% |

It can be seen from Table 2 that the fused set of time and frequency domain transforms had the best performance and the best WA and UA, which is superior to the single feature combination and the original picture form, and also superior to the CNN model with PCA structure. It can be determined that the current organization form of expert 1 can maximize the recognition accuracy of the spectrogram.

Experiment 2: Mainly used for testing expert 2, with 21,517 samples + 513-dimensional + single-layer single-directional GRU model as the baseline; feature information is obtained by using the mean processing method.

Model 1: 2 × 513 dimensions (combined mean and mean square error) + single-layer single-directional GRU model. Model 2: 513-dimensional + single-layer single-directional GRU model + attention mechanism. Model 3: 513-dimensional + double-layer single-directional GRU model + attention mechanism. Model 4: Expert 2, 513-dimensional + three-layer bi-directional GRU model + attention mechanism. The accuracy of each model for emotion recognition was calculated separately.

Table 3 shows the accuracy of speech emotion recognition models after experimental verification.

**Table 3.** Test results of expert 2.

| Model/Sample Set | WA | UA |
|---|---|---|
| Baseline: 513-dimensional + single-layer single-directional GRU model | 52% | 53% |
| Model 1: 2 × 513-dimensional + single-layer single-directional GRU model | 48% | 49% |
| Model 2: 513-dimensional + single-layer single-directional GRU model + attention mechanism | 57% | 58% |
| Model 3: 513-dimensional + single-layer bi-directional GRU model + attention mechanism | 60% | 62% |
| Current System: Expert 2, three-layer single-directional GRU model + attention mechanism | 72% | 71% |

It can be seen from Table 3 that because expert 2 combines the three-layer bi-directional GRU model and attention mechanism, it has the best WA and UA, so it performs best, surpassing the single-layer single-directional GRU model and the model that does not use the attention mechanism. However, it should be noted that after increasing the mean square error characteristic, the overall performance of the model is degraded. It can be determined that the current organization form of expert 2 can maximize the recognition accuracy of the comprehensive information in the local data of the spectrogram, but it is necessary to carefully select a part of the effective integration method.

Experiment 3: Mainly used for testing expert 3. Feature sets that have LLD and HSF characteristics are added to expert 3, then the validity of the feature sets is confirmed. The baseline for this selection is a combination of the INTERSPEECH 2010 Paralinguistic Challenge feature set [25] and the LSTM model.

The feature set contains 1582 features derived from more than 70 LLDs profile values using 20 HSFs. Due to its wide coverage of LLDs and the comprehensiveness of HSF applications, significant effects can be observed with this feature set as the baseline.

Model 1: All spectral features + HSFs (714 dimensions). Model 2: All spectral features (43 dimensions). Model 3: Expert 3's HSFs (20 dimensions). The models are all LSTM, and calculate the WA and UA of emotion recognition. The results are shown in Table 4.

**Table 4.** Test result of expert 3's high-level statistics functionals (HSF) set.

| Model/Sample Set | WA | UA |
|---|---|---|
| Baseline: INTERSPEECH 2010 | 46% | 48% |
| Model 1: 714 dimensions | 48% | 46% |
| Model 2: 43 dimensions | 49% | 47% |
| Current System: Expert 3, 20 dimensions | 58% | 56% |

As can be seen from Table 4, the sample set proposed in model 3 performs the best out of all models, with optimal WA and UA, followed by model 2, and the baseline has the lowest accuracy. From this, it can be determined that the currently selected global feature set can improve the recognition accuracy.

In addition, the multi-level model of expert 3 was compared with experiments with various models. The 1024-dimensional CNN + BLSTM model was used as the baseline.

Model 1: Level 1, 1024-dimensional CRNN + BLSTM. Model 2: Level 2, 20-dimensional HSF + BLSTM. Model 3: 1044-dimensional CRNN + HSF + BLSTM. Model 4: Expert 3, multi-level combined with attention mechanism. The accuracy of each model for emotion recognition was

calculated separately. Table 5 shows the accuracy of speech emotion recognition models after experimental verification.

**Table 5.** Test result of expert 3.

| Model/Sample Set | WA | UA |
|---|---|---|
| Baseline: CNN + BLSTM | 54% | 53% |
| Model 1: Level 1 | 61% | 62% |
| Model 2: Level 2 | 57% | 58% |
| Model 3: CRNN + HSF + BLSTM | 68% | 68% |
| Current System: Expert 3, multi-level | 72% | 71% |

It can be seen from Table 5 that because expert 3 combines the feature set of LLD and HSF and assists the CRNN model to obtain the relevant features of time series, it has the best WA and UA, so it performs best, followed by model 3, which does not use the attention mechanism. The current organization form of expert 3 can use global features and timing signals in the process of emotion recognition.

### 4.3. Experiment with Ensemble Learning Model

From the above experiments, it can be judged that every expert has a positive influence on the accuracy of emotion recognition at their respective angles. For the corpus of sample size balance and data stability, each designed expert has a certain degree of credibility and can complete the emotion recognition task independently. However, after the data are preprocessed, the sample collection often has data skew and irregularities. At this point, the decision of the individual expert may be biased.

In the IEMOCAP corpus, the sample size of the existing angry class is low, while that of the happy class is high. Each expert performs differently in this particular situation when making decisions. At this time, according to the ensemble learning scheme, the confidence of each type of emotion of each expert in a special scene can be calculated, thereby improving the recognition accuracy of each type of emotion through the weighting effect. The effect of the model proposed in this paper is measured from the perspective of experimental comparison before and after the ensemble learning and the confidence comparison of several popular models.

Experiment 4: Mainly used to test the improvement effect of the model before and after ensemble learning, with a single best-performing expert 3 model as the baseline. Model 1: Three experts + confidence weighting calculations. Model 2: Three experts + confidence calculations (using the first integration scheme). Model 4: Current system, three experts + confidence calculations (using the second integration scheme). Each expert's confidence in each emotion recognition was calculated separately. Table 6 shows the accuracy of speech emotion recognition models after experimental verification.

**Table 6.** Test results of the ensemble model.

| Model/Sample Set | WA | UA |
|---|---|---|
| Baseline: Expert 3 | 70% | 70% |
| Model 1: Three experts + confidence weighting | 69% | 68% |
| Model 2: Three experts + confidence weighting (using the first option) | 73% | 72% |
| Current system: Three experts + confidence weighting (using the second option) | 75% | 75% |

Experiment 5: Mainly used to compare the effect of the current ensemble learning model and other popular emotion recognition models. Model 1: Emotion LLDs combined with attention mechanism [26]. Model 2: Fully convolutional network combined with attention mechanism [27]. Model 3: DNN on spectrograms [28]. Current system: ensemble model (using the second option). Each expert's confidence in emotion recognition was calculated separately. Table 7 shows the accuracy of speech emotion recognition models after experimental verification.

**Table 7.** Comparison of ensemble model with other models.

| Model/Sample Set | WA | UA |
|---|---|---|
| Model 1: Emotion LLDS + Attention | 63.5% | 58.8% |
| Model 2: Fully convolutional network + attention | 70.4% | 63.9% |
| Model 3: Spectrograms + DNN | 71.45% | 64.22% |
| Current System: Ensemble model | 75% | 75% |

It can be seen from Table 6 that since the ensemble learning model combines the results of multiple experts, every expert fully utilized their own respective advantages. Therefore, the current system has the best WA and UA, exceeding the recognition accuracy of other ensemble learning models.

Table 7 lists the results of some common emotion recognition models. By comparison, the WA and UA of the current system are better than those of other models.

In addition, through information statistics, it is found that the number of emotions that the third expert guessed wrong and the first two experts answers correct is 446, and the ratio is 0.1; the number of emotions that the third expert answered correctly but the first two experts guessed wrong is 298, and the ratio is 0.06.

From this, it can be determined that the first two experts can correct the wrong decision of the third expert through collective decision-making. The current ensemble learning model can improve the accuracy of emotion recognition in terms of global information and local features for each sentiment category.

Table 8 is the confusion matrix for the specific sentiment categories in the current ensemble learning model. According to the results, recognition accuracy is higher for emotions with higher arousal, such as happy, angry, etc., and lower for categories with lower arousal, such as calmness and sadness.

**Table 8.** Confusion matrix of emotional categories.

| Recognition Accuracy | Happy | Angry | Neutral | Sad |
|---|---|---|---|---|
| Happy | 77.6% | 13% | 5.6% | 3.7% |
| Angry | 3.8% | 73.7% | 14.7% | 7.7% |
| Neutral | 3.4% | 4.4% | 89.2% | 3% |
| Sad | 6.4% | 13.3% | 18% | 62.3% |

Based on the discussion of the above experimental results, the following conclusions can be obtained:

(1) The three weak classifiers (experts) proposed in this paper show their performance ability from different perspectives. From the design point of view, the experts have strong independence and can complement each other. Expert 1 focuses on local features of the spectrogram. Expert 2 uses the local statistical features and the RNN model to emphasize the timing of speech. Expert 3 emphasizes the overall manual features and adds a new feature representation generated by a deep learning model. Through the ensemble of the three experts, the comprehensive description of speech features is reflected.

(2)     Compared with the previously reported single in-depth learning model [29], the ensemble learning model can effectively reduce the negative impact of data imbalance on accuracy, so as to improve the result of WA and some classifications of UA.

(3)     Compared with the single deep learning model, the ensemble learning model can reduce the difference of emotion in valence activation.

(4)     This model has generalization ability, mainly reflected in its input corpus. The IEMOCAP corpus used in the experiment has a certain degree of complexity, including gender, speaker performance status, adult age distribution, background environment, and other factors. This model was applied to complex scenes and had better accuracy, which proves that it has certain generalization ability. In the future, the data provided by the corpus will be further expanded, and the age of the audience will be reduced, and the validity of this model in special populations will be further tested.

## 5. Conclusions

Identifying specific emotions from speech is a challenging task, and the results often depend on the accuracy of the speech signal characteristics and the validity of the model. A multi-view, multi-level ensemble learning model with certain generalization ability was designed with global and local features. Related research was carried out on multi-dimensional speech emotion feature extraction, multi-level neural network model design, and ensemble learning program design. Addressing the data imbalance problem of the IEMOCAP corpus, a confidence calculation method is proposed. To give full play to the role of each expert from different angles, combined with the attention mechanism, the significant areas of important local features are found. Through verification, the proposed model is shown to have higher recognition accuracy and stronger generalization ability.

In future research, a personalized network model structure will be designed from the perspective of the speaker and a general network structure will be sought, which combines personalized features to achieve improved efficiency of speech emotion recognition.

**Author Contributions:** Conceptualization, C.Z. and C.W.; methodology, C.Z. and N.J.; software, C.Z. and N.J.; validation, C.Z. and N.J.; investigation, C.Z. and N.J.; writing—original draft preparation, C.Z. and N.J.; writing—review and editing, C.Z. and N.J.; visualization, C.Z. and N.J.; supervision, C.Z. and N.J. All authors have read and agreed to the published version of the manuscript.

**Funding:** This research was funded by the Natural Science Foundation of Liaoning Province, grant number 20180551068; the National Natural Science Foundation of China, grant number 61370070; the National Natural Science Foundation of China, grant number 61976032; the National Natural Science Foundation of China, grant number 61976124.

**Acknowledgments:** This paper is funded by the Natural Science Foundation of Liaoning Province, Research on Emotional Analysis and Evaluation Model of Speech Reading Based on Machine Learning (20180551068).

**Conflicts of Interest:** The authors declare no conflict of interest.

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
