# Peer review of "An Ensemble Model for Multi-Level Speech Emotion Recognition"

_applsci, doi:10.3390/app10010205_

Round 1
Reviewer 1 Report
This paper submitted by C. Zheng et al. presents an ensemble-based deep neural network model for enhancing the capability of the speech emotion recognition, which is quite an interesting research topic. The paper contains adequate relevant references and the schemes for the proposed methodology are clearly structured. The suggested approach achieves feasible result for improving its performance in respects of accuracy and generalization. However, some parts remains to be revised and specified as written below.
- There is a deficiency of the theoretical background and fundamentals for explaining the proposed method. It is required that the authors should specify theoretically the techniques or the methodologies used for establishing the suggested methods, which are denoted as “Expert” by the authors.
- A more comprehensive discussion in terms of “generalization” should be presented. Please separate the discussion section from the conclusion part and provide the related remarks.
- It is strongly recommended for the authors to raise the quality of the figures and the equations used in the manuscript. Contents of the figures, e.g., illustrations, equations, and their resolutions, should be modified, while the notations of the equations has to be unified.
- Moderate grammar and spell check are required.
Author Response
Dear reviewer:
I am very grateful to your comments for the manuscript. Based on your comment and request, we have made extensive modification on the original manuscript. A revised manuscript with the correction sections by using the "Track Changes" function was attached as the supplemental material.
Some of your questions were answered below.
Comment 1: There is a deficiency of the theoretical background and fundamentals for explaining the proposed method. It is required that the authors should specify theoretically the techniques or the methodologies used for establishing the suggested methods, which are denoted as “Expert” by the authors.
The authors’ Answer: Thanks for the referee’s kind advice. Just like what the referee said, In Section 3.1 of the paper, the author adds the basic design principles and technical features of three experts. In summary, the basic ideas of the three experts are as follows:
Expert 1 uses the spectrogram which is a kind of three-dimensional spectrum, which contains rich time and frequency information. The expert choose the CNN model that can highlight the expression of local features.
Expert 2 uses the characteristics of short-term invariance of speech. Speech is a sequential sequence, so it is suitable for GRU, which is a deep model of time series.
Expert 3 uses LLD features and its HSF can describe features from a global point of view, and can be used as a powerful supplement to existing time series features. The abstract representation of advanced features obtained from deep learning model is generated.
Comment 2: A more comprehensive discussion in terms of “generalization” should be presented. Please separate the discussion section from the conclusion part and provide the related remarks.
The authors’ Answer: Thanks for the referee’s suggestion. In the revised manuscript, the author separated the discussion part from the conclusion part. In the fourth section of the thesis, the author added four key discussions including “generalization”(the 4th part). A brief summary is as follows. For details, please refer to the revised version in the attachment.
The three weak classifiers (experts) proposed in this paper show their performance ability from different perspectives. From the design point of view, experts have strong independence, and they can complement each other. Compared with the single in-depth learning model published by the author before, using the ensemble learning model can effectively reduce the negative impact of data imbalance on accuracy, so as to improve the result of WA. The ensemble learning model can reduce the difference of emotion in valence activation. This model has the ability of generalization, mainly reflected in its input corpus. This model is applied to this kind of complex scene and gets better accuracy, which proves that this model has certain generalization ability. In the future, the data provided by the corpus will be further expanded, and the age of the audience will be reduced, the validity of this model in special population will be further tested.Comment 3: It is strongly recommended for the authors to raise the quality of the figures and the equations used in the manuscript. Contents of the figures, e.g., illustrations, equations, and their resolutions, should be modified, while the notations of the equations has to be unified.
The authors’ Answer: Thanks for the referee’s good evaluation and kind suggestion. The author has unified the symbols in the formula, modified formulas 1, 4 and 6, and unified the symbol expression of the integrated learning formula.
In addition, the author also adjusted the presentation and layout of Fig 1. The description of the form number is unified. For details, please refer to the revised version in the attachment.
Comment 4: Moderate grammar and spell check are required.
The authors’ Answer: Thanks for the referee’s suggestion. The author intends to adopt the English editing service providing by MDPI to check the grammar and spelling. In the revised manuscript, many grammatical or typographical errors have been revised. All the marks indicated above are in the revised manuscript.
Thank you and all the reviewers for the kind advice.
Should you have any questions, please contact us without hesitate.

Reviewer 2 Report
The paper is generally well written with very detailed descriptions. However, in the beginning of the paper, it is not clear whether the 3 experts are human experts or the results of 3 different approaches/algorithms.
In Expert 1 scenario, why "the time domain and the frequency domain information of the spectrogram are flipped at various angles"?
Also in Expert 2 scenario, I didn't understand on which data the PCA operation was performed ("original spectral data is selected, the time domain and frequency domain information are retained, and the depth information in the map is discarded, and the PCA operation is performed.".
Authors' previous paper "Research on Speech Emotional Feature Extraction Based on Multidimensional Feature Fusion" must be cited in this work and original contribution of this article must be clearly stated.
Author Response
Dear reviewer:
I am very grateful to your comments for the manuscript. Based on your comment and request, we have made extensive modification on the original manuscript. A revised manuscript with the correction sections by using the "Track Changes" function was attached as the supplemental material.
Some of your questions were answered below.
Comment 1: The paper is generally well written with very detailed descriptions. However, in the beginning of the paper, it is not clear whether the 3 experts are human experts or the results of 3 different approaches/algorithms.
The authors’ Answer: Thanks for the referee’s kind advice. In 1 introduction, the author added an explanation for three experts. In order to avoid misunderstandings, we usually use the classifier: expert.
Comment 2: In Expert 1 scenario, why "the time domain and the frequency domain information of the spectrogram are flipped at various angles"?
The authors’ Answer: Thanks for the referee’s suggestion. In Section 3.2, the author added a detailed description of this operation. There are two purposes of this kind of multi angle flipping: (1) data enhancement; (2) the generalization ability of speech features extracted by CNN network model is stronger.
Comment 3: Also in Expert 2 scenario, I didn't understand on which data the PCA operation was performed ("original spectral data is selected, the time domain and frequency domain information are retained, and the depth information in the map is discarded, and the PCA operation is performed.".
The authors’ Answer: Thanks for the referee’s good evaluation and kind suggestion. For all the original audio data after segmentation, the author performs time-domain and frequency-domain calculation, and then performs PCA operation, which plays a role of dimension reduction to a certain extent.
Comment 4: Authors' previous paper "Research on Speech Emotional Feature Extraction Based on Multidimensional Feature Fusion" must be cited in this work and original contribution of this article must be clearly stated.
The authors’ Answer: Thanks for the referee’s suggestion. Just like what the referee said, Compared with previous papers, this paper mainly has innovative research in the following aspects:
(1)Features: following the part of LLD mentioned in the paper, we added spectrogram, transformed spectrogram and multi feature expression from deep learning model.
(2)Model: CRNN model mentioned in the paper is used as the second channel of the third expert, and CNN, GRU, MLSTM, ensemble learning and other models are added.
Through the improvement of the above content, the author designed a reasonable integrated learning model, using the weak classifier to learn the voting mechanism, and finally improved the overall WA and part of the emotional UA, in order to obtain better emotional recognition effect.
In the analysis part of the experimental results of the manuscript, the comparative analysis of the experimental results of two papers is added.
In the revised manuscript, many grammatical or typographical errors have been revised. All the marks indicated above are in the revised manuscript.
Thank you and all the reviewers for the kind advice.
Should you have any questions, please contact us without hesitate.
